# Chemical Differentiation and Quantitative Analysis of Black Ginseng Based on an LC-MS Combined with Multivariate Statistical Analysis Approach

**DOI:** 10.3390/molecules28135251

**Published:** 2023-07-06

**Authors:** Lele Li, Zhixia Chang, Keyu Wei, Yi Tang, Zhao Chen, Hongli Zhang, Yang Wang, Heyun Zhu, Bo Feng

**Affiliations:** 1School of Pharmacy, Jilin Medical University, Jilin 132013, China; lilele19911006@163.com (L.L.); m17693201684@163.com (Z.C.); 18776236716@163.com (K.W.); m18883784475@163.com (Y.T.);; 2Jilin Ginseng Academy, Changchun University of Chinese Medicine, Changchun 130117, China

**Keywords:** black ginseng, ginsenosides, UPLC-Q-TOF/MS, processing, multivariate statistical analysis

## Abstract

Black ginseng is a new type of processed ginseng that is traditionally used in herbal medicine in East Asian countries. It is prepared from fresh, white, or red ginseng by undergoing a process of steaming and drying several times. However, the chemical differentiation of black ginseng with different processing levels is not well understood. The aim of this study was to propose a new method for discriminating and quantifying black ginseng. Six ginsenosides from black ginseng were accurately quantified, and based on this, the black ginseng samples were divided into incomplete and complete black ginseng. Ultrahigh-performance liquid chromatography–quadrupole-time of flight/mass spectrometry (UPLC-Q-TOF/MS) combined with a multivariate statistical analysis strategy was then employed to differentiate the two groups. A total of 141 ions were selected as analytical markers of black ginseng, with 45 of these markers being annotated by matching precise *m*/*z* and MS/MS data from prior studies.

## 1. Introduction

Ginseng is the roots and rhizomes of *Panax ginseng* C.A. Meyer and has been widely used as an herbal remedy or tonic food in China, Korea, and Japan for thousands of years to adjust the balance of the human body [1,2,3]. Processing plays a crucial role in the utilization of herbal medicines in traditional Chinese medicine. In the commercial market, ginseng is commonly sold in two forms: white and red ginseng. The white ginseng is produced by drying fresh ginseng in the sun after basic cleaning, and the red ginseng is manufactured by steaming fresh ginseng one time. Traditionally, white ginseng and red ginseng are generally used. Black ginseng is a new type of processed ginseng which exhibits more potent biological activities than the two traditional processed products [4]. Black ginseng is also available and is typically produced by steaming fresh or white ginseng at 96 °C for three hours, followed by hot-air-drying and then repeating the above operation nine times [5,6,7]. Other methods have also been reported in the preparation of black ginseng, such as steaming the white ginseng at 113.04 °C for 18 h and drying at 100 °C for 8.03 h [8]. According to previous reports, black ginseng exhibits anticancer, hepatoprotective, antidiabetic, anti-obesity, antioxidant, and anti-inflammatory pharmacological activities [9,10,11,12,13,14].

Some reports indicated that the main bioactive secondary metabolites of ginseng are ginsenosides [15,16]. With respect to the structural characteristics of aglycone, ginsenosides can be classified into three major classes: protopanaxadiols (PPDs), protopanaxatriols (PPTs), and the oleanolic acid type [17]. In addition, other classes have been reported in ginseng, such as malonyl-ginsenoside and acetyl-ginsenoside [18,19]. During the steaming process of black ginseng, certain ginsenosides undergo various chemical reactions and are converted into different compounds. Specifically, the polar ginsenosides undergo hydrolysis, dehydration, decarboxylation, and isomerization reactions at C-3, C-6, or C-20, resulting in the formation of specific less-polar ginsenosides [7,20]. Some works have been reported and aimed to study the transformation of ginsenosides [4,21,22,23]. The changes in the chemical composition of black ginseng will significantly affect its biological activity, pharmacological effects, and clinical applications. However, relatively less effort has been devoted to studying the chemical differentiation between black ginseng with different processing levels. Therefore, there is an urgent need to develop a quantitative analysis method and explore analytical markers of black ginseng, as such actions are beneficial for the quality control of black ginseng.

Innovative approaches utilizing LC-MS and multivariate statistical analysis (MSA) have proven successful in exploring analytical markers and evaluating the quality of traditional Chinese herbal medicine [24]. In this study, an ultrahigh-performance liquid chromatography–triple quadrupole/mass spectrometry (UPLC-QQQ/MS) method was developed to precisely quantify six ginsenosides in white and black ginseng. Based on the quantitative results, the black ginseng samples were categorized into two groups: incomplete and complete black ginseng. Subsequently, the ultrahigh-performance liquid chromatography–quadrupole-time of flight/mass spectrometry (UPLC-Q-TOF/MS) technique was employed in conjunction with MSA to investigate the differences between the two groups and select the analytical markers of black ginseng.

## 2. Results and Discussions

### 2.1. Optimization of UPLC-QQQ/MS Conditions

Figure 1A displays the chromatogram of six ginsenoside standards obtained through UPLC-QQQ/MS, utilizing the MRM mode. The ion pairs utilized for quantitative analysis were optimized. Typically, the precursor ion was the base peak in the MS spectra of a compound, while the product ion was selected as the base peak in the MS/MS spectra. Table 1 showcases all the optimized MRM conditions. The precursor ions were [M − H]^−^ or [M + HCOO]^−^ of ginsenosides, while the selected product ions were [M − H]^−^ of ginsenosides or fragment ions from losing sugar radicals. The optimized collision energy was from 15 to 35 eV. High collision energy would cause the precursor ions to fragment into smaller fragment ions, resulting in a decrease in the signal of the target product ions. Therefore, the best collision voltage used here does not exceed 35 eV. Suitable declustering potential is beneficial to the generation of adduct ions (precursor ions) of ginsenoside, and the optimized declustering potential was from −80 to −130 V. As shown in Figure 1B,C, all the optimized parameters were used in the UPLC-QQQ/MS analysis to quantify the six ginsenosides in white and black ginseng.

### 2.2. Validation of the Method

Table 2 displays the precision and repeatability of the developed UPLC-QQQ/MS method for quantifying ginsenosides in black ginseng. The intra-day precisions for all six ginsenosides ranged from 0.90% to 1.86%, while the repeatability ranged from 1.33% to 3.74%. The LODs and LOQs for the six ginsenosides were in the ranges from 4.0 to 20.4 ng/mL and 8.0 to 51.0 ng/mL, respectively. The regression equation and linear range of ginsenosides are also displayed in Table 2. The six ginsenosides showed a wide linear range, with Rh2 and Rg3 having a linear range that is 500-times wider. The compounds exhibited good linearity (r ≥ 0.99) and a concentration range of up to two orders of magnitude, indicating the method’s capability to quantify the ginsenosides. The recoveries ranged from 86.16% to 112.39%, as presented in Table 2. The validation results of the method indicated that the developed UPLC-QQQ/MS method has low LODs and LOQs, excellent intra-day precision, repeatability, and accuracy. This method can be effectively utilized to quantify the six ginsenosides present in black ginseng samples.

### 2.3. Content Determination of Six Ginsenosides in Black Ginseng

The content of six ginsenosides in white and black ginseng was detected using the developed UPLC-QQQ/MS method. The quantitative results of six ginsenosides were shown in Table 3, and the results illustrate that the content of all the six ginsenosides changed significantly during the processing of black ginseng. The content of polar ginsenoside Re, Rg1, and Rb1 decreased significantly with the increase in processing times and began to stabilize from the seventh processing. On the contrary, the content of less-polar ginsenoside Rh1, Rh2, and Rg3 increased significantly with the increase in processing times and changed a little after the seventh processing. Therefore, the processing of black ginseng tends to finish after seven cycles of steaming and drying. The black ginseng samples were divided into two groups in the next section: one group is incomplete black ginseng (less than 7 cycles of steaming and drying), and the other group is complete black ginseng (7 or more than 7 cycles of steaming and drying). According to the report, the common chemical change of ginsenosides is the predominance of Rg3, and the Rg3 content of complete black ginseng is more than 1 mg/g [4]. Our experimental results were consistent with the abovementioned literature. The black ginseng samples which were processed through 7 cycles of steaming and drying conformed to the standard. In addition, the content of Rg3 in all complete black ginseng (7 or more than 7 cycles of steaming and drying) exceeded 1 mg/g. The results indirectly supported the classification of black ginseng samples into two groups based on the number of times they were processed, namely incomplete and complete black ginseng.

### 2.4. Exploring of Chemical Markers of Black Ginseng Based on Multivariate Statistical Analysis

The established UPLC-Q-TOF/MS method was used to compare the chemical profiles of white and black ginseng samples (Figure 2). Differences between incomplete and complete black ginseng were observed in their total ion chromatograms (TIC). Compared with the incomplete black ginseng, the relative height of the chromatographic peak of ginsenosides with short retention time and high polarity in complete black ginseng was decreased, while the relative height of the chromatographic peak of ginsenosides with long retention time and low polarity was increased. MSA was applied to display intergroup differences. After peak extraction, a dataset containing information on 309 ions was generated. An OPLS-DA pattern was built using the data of black ginseng samples. As shown in Figure 3A, all samples were within the Hotelling T2 (0.95) ellipse, and distinct differentiation between incomplete and complete black ginseng suggests variations in constituent composition.

To identify the key chemical components responsible for distinguishing between incomplete and complete black ginseng, the OPLS-DA was used to generate a loading plot (Figure 3B) and VIP plot (Figure 3C). The loading plot displays each detected component in the black ginseng samples, with those further from the origin contributing more strongly to discrimination. The features with a VIP1 large than 1 were selected initially as important analytical markers, which were then further screened using Student’s *t*-test to identify components with significant differences between the two groups. Ultimately, 141 ions with *p* < 0.05 were kept as the analytical markers, which are highlighted in red in the VIP plot.

Analytical markers were identified by matching accurate *m*/*z* and MS/MS information from standard references or the literature [15,18,19]. Quinquenoside R1, tentatively annotated without a standard reference, was used as an example to demonstrate identification procedures. The [M + HCOO]^−^ ion of Quinquenoside R1 at *m*/*z* 1195.61 was selected as a precursor ion. As shown in Figure 4A, the product ion with the highest signal intensity (*m*/*z* 1149.60) results from the loss of HCOOH, and product ions at *m*/*z* 1107.59, 1089.58, 987.54, 945.55, 927.53, and 783.43 correspond to the loss of acetyl or glucose radicals. Except for the ion *m*/*z* 987.54, all the other fragment ions are products obtained from the loss of acetyl from the precursor ion, and the possible fragmentation pathways for the product ions are shown in Figure 4B. A total of 45 analytical markers were tentatively annotated by examining their MS spectra and MS/MS fragmentation patterns, as summarized in Table 4. As shown in Table 4, during the process of processing incomplete black ginseng into complete black ginseng, 25 of the 45 analytical markers showed an increase in content, while 20 compounds showed a decrease in content. The retention times of the 25 compounds with increased content were in a range from 4.40 to 14.75 min, with a molecular weight range of 460.3916–908.4981 Da, while the retention times of the 20 ginsenosides with decreased relative content were in a range from 4.41 to 10.91 min, with a molecular weight range of 800.4922–1240.6452 Da. Overall, compared to ginsenosides with decreased content, ginsenosides with increased content generally had a longer retention time and smaller molecular weight, showing smaller polarity. The changes in ginsenosides of black ginseng significantly affect its biological activity and pharmacological effects. Our experimental results partially explain why black ginseng has better pharmacological effects, including anticancer, hepatoprotective, antidiabetic, anti-obesity, antioxidant, and anti-inflammatory effects [9,10,13,25,26,27].

Quinquenoside R1 is an acetylated ginsenoside, and the variation of its content in processing is different from other polar or less-polar ginsenosides. As shown in Figure 4C, the intensity of quinquenoside R1 increased significantly during the first steaming and drying. Then, the intensity of quinquenoside R1 decreased from the second cycle to the last one. The quinquenoside R1 could result from the fact that ginsenoside mRb1 lost CO_2_ at its malonyl, which could explain the increase of quinquenoside R1 during the first cycle. However, as a polar ginsenoside, the quinquenoside R1 transformed into specific less-polar ginsenosides by hydrolysis, dehydration, decarboxylation, and isomerization reactions during the cycles of steaming and drying. Under the combined effect of the aforementioned factors, the variation of quinquenoside R1 content is different from other polar or less-polar ginsenosides. In addition, the variation of some other acetylated ginsenosides, including 6′-*O*-acetyl-Rg1, 6′-*O*-acetyl-Re, yesanchinoside D, Rs1, and Rs2, is similar to quinquenoside R1 during the processing of black ginseng. In addition to ginsenosides, the processing also affects the intensity of aglycones. As illustrated in Figure 5, the intensity of both 20(S)-protopanaxadiol (PPD) and 20(S)-protopanaxatriol (PPT) increased during the steaming and drying cycles. PPD and PPT are the final metabolites of protopanaxadiol-type and protopanaxatriol-type ginsenosides, respectively, and have a broad spectrum of bioactive effects on the human body [28,29].

### 2.5. Brief Summary

Overall, during the processing of black ginseng, the polar ginsenosides were transformed into less-polar ginsenosides. Protopanaxanediol saponins Ra1, Ra2, Ra3, Rb1, Rb2, Rc, and Rd underwent hydrolysis at the sugar moiety on positions C-3 and C-20 to produce Rg3, F2, and Rh2 [30,31,32]. Hydrolysis and dehydration of Rg1, Re, and Rg2 generated Rh1, Rh3, and others [33,34]. Quinquenoside R1, 6′-*O*-acetyl-Rg1, and 6′-*O*-acetyl-Re were decarboxylated from mRb1, mRg1, and mRe, respectively, while these acetylated saponins could further hydrolyze to produce Rg3, F2, Rh1, Rh2, Rh3, and others [20]. PPT and PPD compounds were produced by the hydrolysis of corresponding ginsenosides. Among the 45 analytical markers, compound Nos. 6, 22, 26, 28, 29, 31, 32, 36, and 41 were not well identified. Other ginsenosides, especially those identified using standard substances, are better markers.

## 3. Materials and Methods

### 3.1. Chemical Reagents and Materials

HPLC-grade acetonitrile, methanol, and formic acid were procured from TEDIA (Fairfield, OH, USA), while ultrapure water was obtained using the Milli-Q water purification system (Millipore, Bedford, MA, USA). The ginsenoside Rb1, ginsenoside Re, ginsenoside Rg1, ginsenoside Rh1, ginsenoside Rh2, and ginsenoside Rg3 standards (abbreviated as Rb1, Re, Rg1, Rh1, Rh2, and Rg3, respectively, in the subsequent sections) were obtained from Chengdu Must Co. (purity: 99% HPLC, Chengdu, China). Fresh ginseng (four-year-old cultivation) was purchased from Wanliang market (Jilin, China) in 2021.

### 3.2. Sample Preparation

Accurate weighing and dissolution of standard references (Rb1, Re, Rg1, Rh1, Rh2, and Rg3) in 70% methanol resulted in the preparation of a stock solution. Dilution of the mixed standards’ stock solutions to different concentrations using 70% methanol facilitated method validation. All solutions were stored at 4 °C after preparation. 

Fresh ginseng was washed and dried to obtain white ginseng. Four batches of black ginseng samples were manufactured by repeating the steaming of white ginseng nine times at 98 °C for 3 h and drying at 50 °C for 24 h. A part of the black ginseng samples was retained after each steaming and drying.

One gram of dried white ginseng and black ginseng samples was grounded into powder with a mortar, and 10 mL of 70% methanol aqueous solution was then added. After being ultrasonic extracted for 60 min at room temperature, the extract was filtered through a syringe filter (0.22 μm) and injected directly into the UPLC system.

### 3.3. Method Validation

A quantitative analysis was conducted using an external calibration method. Linear calibration curves were generated by plotting the peak areas of mixed standard solutions against their corresponding concentrations. The limit of detection (LOD) and limit of quantification (LOQ) were determined using a signal-to-noise ratio of 3 and 10, respectively. Intra-day precision was evaluated by analyzing the standard solution six times and calculating the relative standard deviation (RSD) of the results. To test repeatability, six independent sample solutions were prepared using the procedures outlined in the previous section. The standard addition method was used to determine the recovery of this method, which was calculated using the following formula:Recovery(%) = (observed amount − original amount)/spiked amount × 100%

### 3.4. UPLC-QQQ/MS Analysis

A Shimadzu LC20AD ProminenceTM UPLC system and a Thermo Fisher Golden C18 column (2.1 × 50 mm, 1.9 μm) maintained at 35 °C was used for chromatographic separation. Then, 0.1% formic acid in water was used as mobile phase A, and 0.1% formic acid in ACN was used as mobile phase B. The proportion of mobile phase B was as follows: 5% (time is 0 to 1 min), 5–30% (time is 1 to 2 min), 30–40% (time is 2 to 10 min), 40–95% (time is 10 to 14 min), and 95–95% (time is 14 to 17 min), followed by a return to 5% at 17.1 min for a 3 min equilibration period. The volume of injection and flow rate were set to 2.0 μL and 0.3 mL/min, respectively.

An AB 3200 MS (SCIEX, Concord, Canada) equipped with an electrospray ionization source was used for QQQ/MS analysis. The MS operated in negative ion mode with the following parameters: source temperature of 500 °C, ion spray voltage of −4500 V, nebulizer gas (N2) at 50 psi, heater gas (N2) at 50 psi, and curtain gas (N2) at 35 psi.

### 3.5. UPLC-Q-TOF/MS Analysis

The chromatographic separation conditions were identical to those described in Section 2.4, with the exception of gradient elution. The mobile phase B was utilized in the following proportions: 15% (time is 0 to 1 min), 15–40% (time is 1 to 10 min), 40–95% (time is 10 to 15 min), 95–95% (time is 15 to 17 min), and 95–5% (time is 17 to 17.1 min), followed by a return to 5% at 17.1 min for a 2.9 min equilibration period.

The Q-TOF/MS detection was performed using a Triple-TOF 5600 + MS (SCIEX, Concord, Canada) with an electrospray ionization source. Parameters of MS were set to negative ion mode, with an electrospray ionization source temperature of 500 °C, an ion spray voltage of −4500 V, the nebulizer gas (N2) at 55 psi, the heater gas (N2) at 60 psi, and the curtain gas (N2) at 35 psi. The declustering potential was set to 100 V. Full-scan MS data were acquired in TOF/MS mode from *m*/*z* 100 to 2000, with a collision energy of 5 eV. MS/MS data were acquired in IDA mode, with a collision energy of 35 eV and a rolling collision energy of 15 eV. The MS spectra mass range was from *m*/*z* 100 to 2000, and the mass range of MS/MS spectra was from *m*/*z* 100 to 2000.

### 3.6. Multivariate Statistical Analysis

After conducting UPLC-Q-TOF/MS detection, the original data obtained from the white ginseng, black ginseng, and QC samples were processed by MS-DIAL version 4.10 URL (https://mtbinfo-team.github.io/mtbinfo.github.io/MS-DIAL/tutorial.html#chapter-1, accessed on 3 March 2023). This included the subtraction of background, detection of components, peak alignment, and ion fusion. Peak alignment was performed using QC data as the reference file, with a retention-time tolerance of 6 s and an MS tolerance of 10 mDa. Parameters of data collection were set at an MS tolerance of 10 mDa and an MS/MS tolerance of 20 mDa. Parameters of peak detection included a minimum peak intensity of 1000 amplitude.

The resulting dataset, which included *m*/*z* values at retention time, normalized intensity, and sample codes, was utilized for MSA. The dataset was saved as .csv files and imported into SIMCA software 13.0 (Umetrics, Umea, Sweden) to perform orthogonal partial-least-squares discriminant analysis (OPLS-DA). Ions with variable importance in projection (VIP) 1 values greater than 1 were highlighted in the OPLS-DA model and further filtered by Student’s *t*-test (SPSS19.0, Chicago, IL, USA). Components with *p* < 0.05 were deemed significant and selected as potential markers in black ginseng.

## 4. Conclusions

A rapid and reliable method was developed using UPLC-QQQ/MS to simultaneously determine six ginsenosides (Rb1, Re, Rg1, Rh1, Rh2, and Rg3) in black ginseng. Additionally, an approach combining UPLC-Q-TOF/MS with MSA was successfully applied to discover chemical markers of black ginseng. A total of 45 analytical markers were selected and identified by matching accurate *m*/*z* and MS/MS information from standard references or the literature; their changes during the processing were also revealed. This approach has great potential for the quality assessment of ginseng processing products, which can be further applied to the analysis of traditional Chinese medicine.

## Figures and Tables

**Figure 1 molecules-28-05251-f001:**
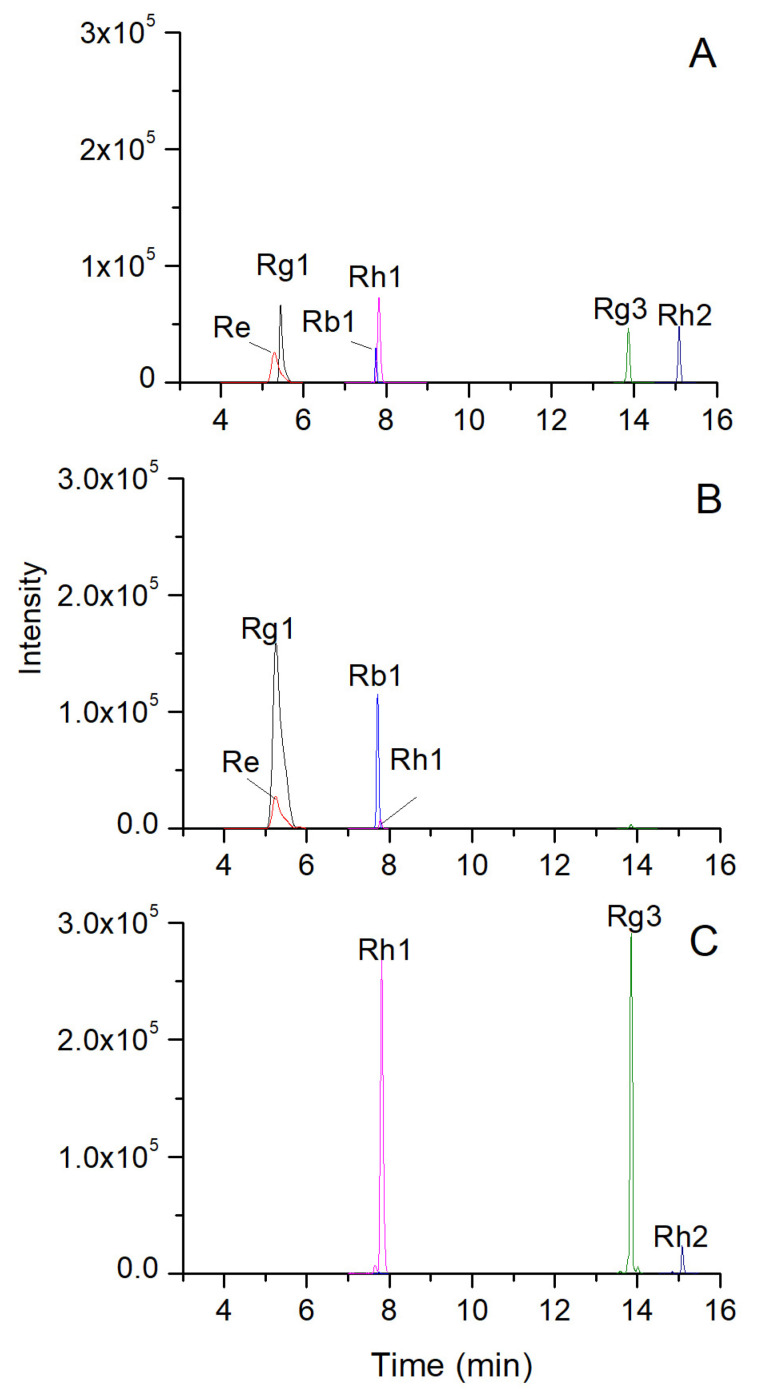
Extract ion chromatograms of the six ginsenosides in mixed standards (**A**), white ginseng (**B**), and black ginseng (**C**) based on MRM mode.

**Figure 2 molecules-28-05251-f002:**
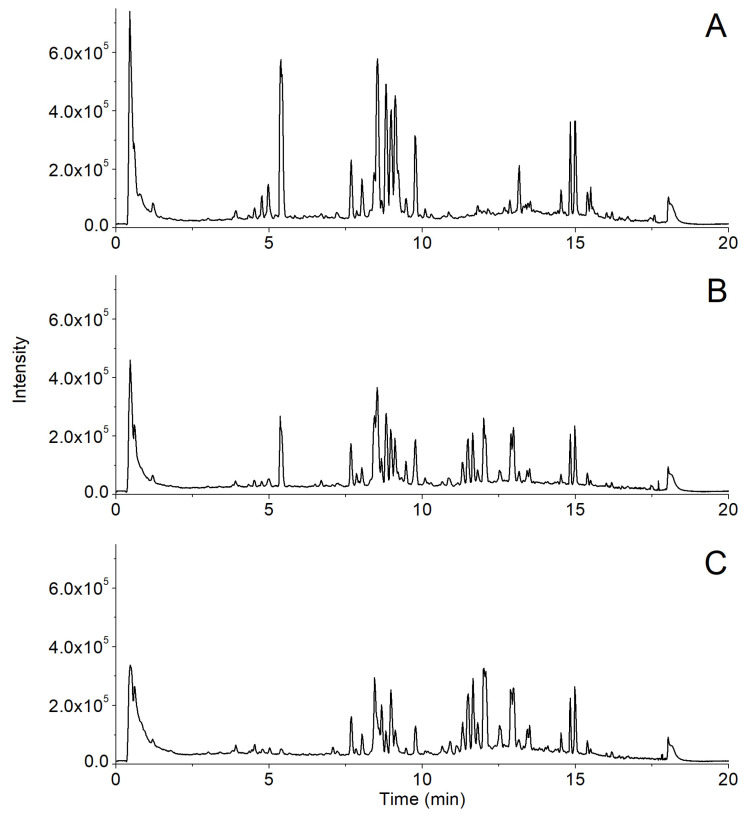
Representative chromatograms of white ginseng (**A**), uncompleted black ginseng (**B**), and completed black ginseng (**C**) samples obtained by UPLC-Q-TOF/MS.

**Figure 3 molecules-28-05251-f003:**
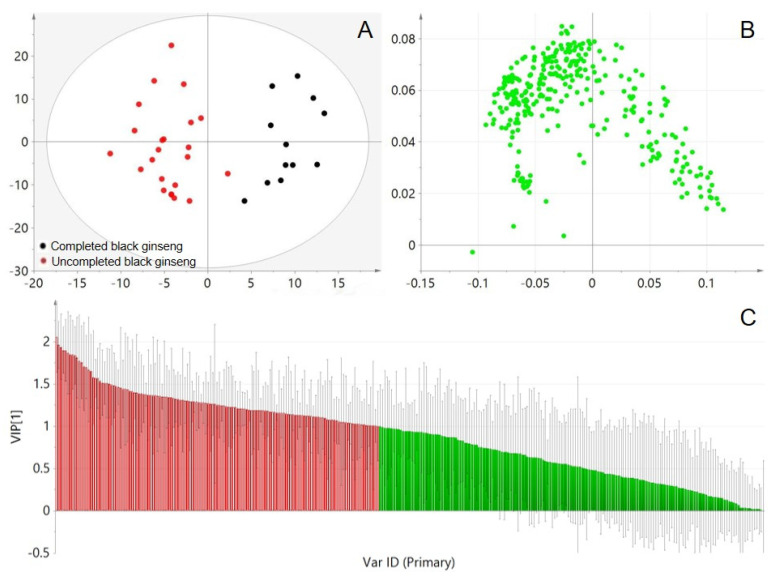
OPLS-DA score plot (**A**) and the corresponding loading plot (**B**) of black ginseng samples. Potential analytical markers (VIP1 > 1, *p* < 0.05) are marked with red color and the other compounds are marked by green color in the loading plot (**C**).

**Figure 4 molecules-28-05251-f004:**
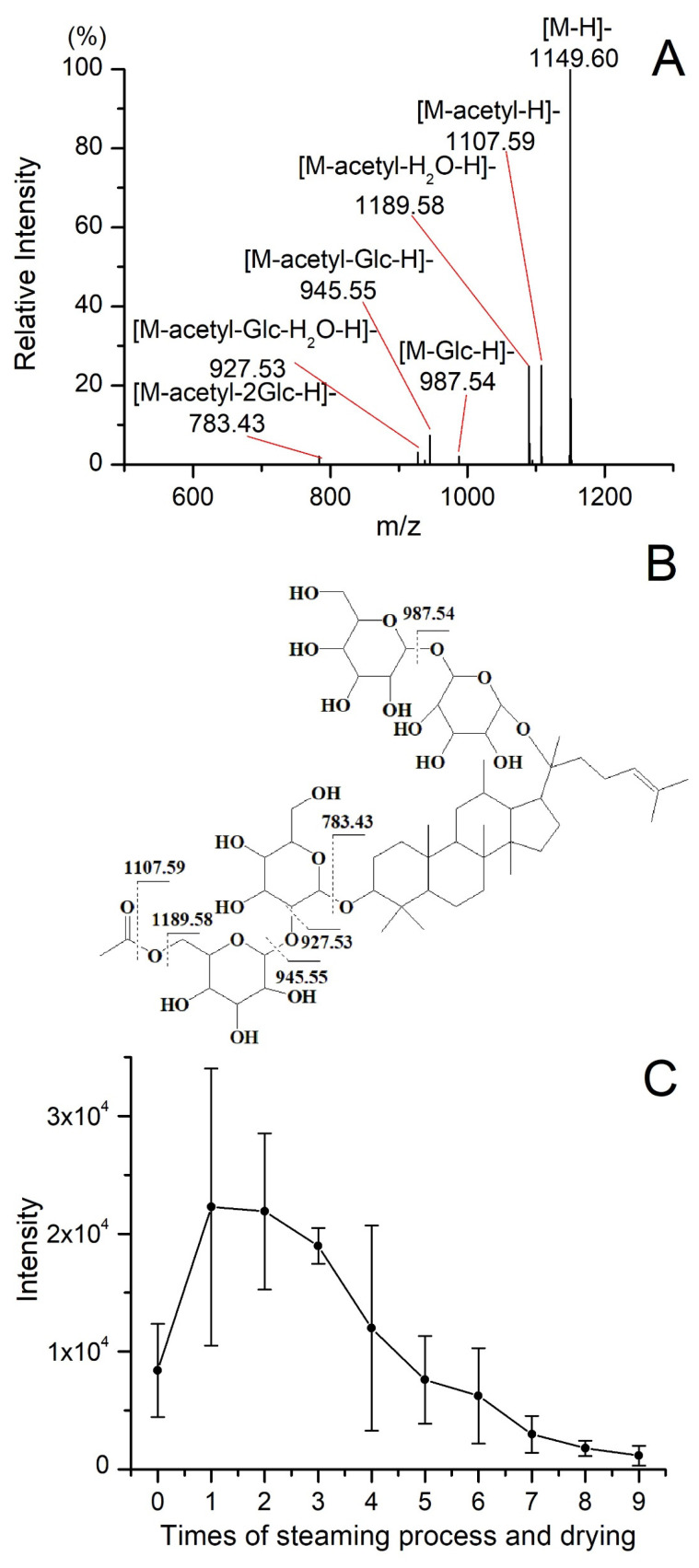
The MS/MS spectrum (**A**) and possible fragmentation pathways of Quinquenoside R1 ion (**B**). The variation of intensity of Quinquenoside R1 during the processing of black ginseng (**C**).

**Figure 5 molecules-28-05251-f005:**
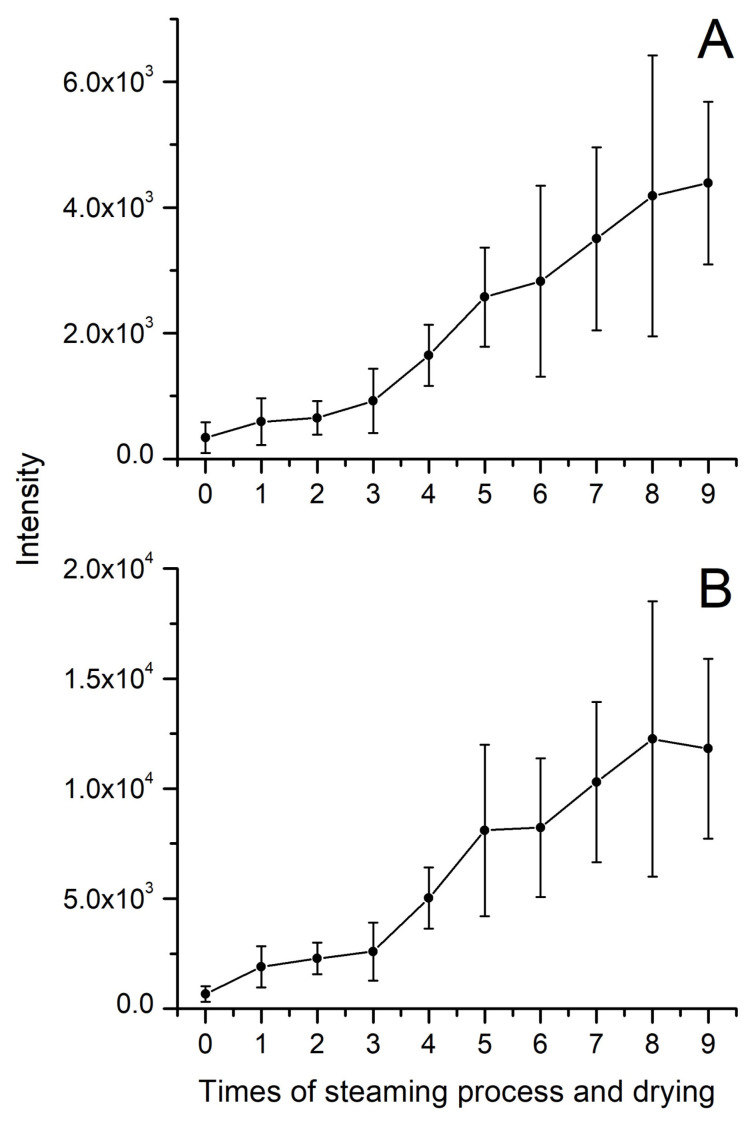
The variation of intensity of protopanaxadiol (**A**) and protopanaxatriol (**B**) during the processing of black ginseng.

**Table 1 molecules-28-05251-t001:** Chemical information of the six ginsenoside standards and results of optimization for MS parameters.

Analyte	Formula	RT (min)	Precursor Ion	Product Ion	Collision Energy (eV)	Declustering Potential (V)
Rg1	C_42_H_72_O_14_	5.20	846	800	15	−80
Re	C_48_H_82_O_18_	5.20	946	638	30	−120
Rb1	C_54_H_92_O_23_	7.76	1108	946	25	−130
Rh1	C_36_H_62_O_9_	7.78	684	638	15	−80
Rg3	C_42_H_72_O_13_	13.87	784	460	35	−120
Rh2	C_36_H_62_O_8_	15.08	668	622	15	−90

**Table 2 molecules-28-05251-t002:** Results of method validation.

Analyte	LOD (ng/mL)	LOQ (ng/mL)	RSD (%) of Intra-Day Precision (*n* = 6)	RSD (%) ofRepeatability (*n* = 6)	Regression Equation	Linear Range (μg/mL)	Correlation Coefficient(*r*)	Recovery (%)
Rg1	11.1	22.2	1.74	1.33	*y* = 81,546*x* − 10,860	0.0222–5.55	0.9993	104.85%
Re	18.2	45.5	1.12	2.86	*y* = 20,558*x* − 4043	0.0455–4.55	0.9955	95.24%
Rb1	20.4	51.0	1.41	3.74	*y* = 17,022*x* − 3209	0.0510–10.20	0.9995	89.31%
Rh1	5.3	10.6	0.90	1.74	*y* = 159,799*x* − 681	0.0106–2.65	0.9998	112.39%
Rg3	4.0	8.0	2.20	2.64	*y* = 98,863*x* + 8345	0.0080–4.00	0.9956	86.16%
Rh2	5.4	10.8	1.86	2.91	*y* = 63,590*x* + 8546	0.0108–5.40	0.9965	99.76%

**Table 3 molecules-28-05251-t003:** Concentration of analytes in black ginseng with different times of steaming process and drying.

Times of Steaming Process and Drying	Concentration of Analytes (μg/g)					
	Re	Rg1	Rb1	Rh1	Rg3	Rh2
0	1599.97 ± 164.67	2651.98 ± 658.15	3201.75 ± 197.14	11.88 ± 0.91	6.361 ± 0.41	0
1	1212.02 ± 55.26	1954.23 ± 630.68	2013.57 ± 131.65	90.28 ± 20.69	226.95 ± 41.18	0
2	704.60 ± 61.62	1726.80 ± 149.11	1769.65 ± 103.98	160.25 ± 41.78	449.94 ± 64.74	0.39 ± 0.27
3	590.59 ± 153.03	595.11 ± 265.37	1365.81 ± 286.76	183.88 ± 67.00	659.84 ± 85.91	4.95 ± 3.48
4	254.12 ± 132.65	591.49 ± 215.37	1023.85 ± 223.20	247.52 ± 68.80	714.56 ± 70.65	11.62 ± 5.55
5	87.07 ± 23.64	151.32 ± 74.67	764.30 ± 179.73	306.75 ± 78.06	895.07 ± 61.20	18.59 ± 4.06
6	53.37 ± 30.78	102.93 ± 67.09	500.37 ± 181.51	339.32 ± 46.78	1071.67 ± 67.74	24.48 ± 8.91
7	17.82 ± 12.41	30.99 ± 25.04	232.25 ± 122.69	369.26 ± 43.33	1202.24 ± 54.82	35.18 ± 9.29
8	9.67 ± 5.89	0	158.19 ± 95.84	373.67 ± 55.53	1376.93 ± 186.82	44.29 ± 18.55
9	7.42 ± 3.14	0	82.35 ± 51.89	376.60 ± 60.01	1343.91 ± 198.74	49.12 ± 15.45

**Table 4 molecules-28-05251-t004:** The detailed information of analytical markers which were identified.

No.	RT (min)	Measured *m*/*z*	Mass Deviation (ppm)	Adducts	Productions	Compounds	Formula	Molecular Weight	Variation
1	4.41	1007.5529	9.63	[M + HCOO]^−^	961.53, 799.48	Re1	C_48_H_82_O_19_	962.545	↓
2	4.44	701.4477	−0.71	[M + HCOO]^−^	655.44, 493.39	494-Glc	C_36_H_64_O_10_	656.4499	↑
3	4.71	847.5099	4.48	[M + HCOO]^−^	801.50, 655.44, 493.39	Rf2	C_42_H_74_O_14_	802.5078	↑
4	5.37	845.4924	2.37	[M + HCOO]^−^	799.49, 637.43, 475.38	Rg1 ^a^	C_42_H_72_O_14_	800.4922	↓
5	5.44	991.5546	6.35	[M + HCOO]^−^	945.54, 799.49, 637.43, 475.38	Re ^a^	C_48_H_82_O_18_	946.5501	↓
6	6.11	699.428	−6.43	[M + HCOO]^−^	653.43, 491.38, 161.04	noto-R8/noto-R9 /M7cd/notopanaxoside-A	C_36_H_62_O_10_	654.4343	↑
7	6.71	887.5053	4.85	[M + HCOO]^−^	841.49, 781.47, 475.38	6′-*O*-acetyl-Rg1	C_44_H_74_O_15_	842.5028	↓
8	7.03	863.499	−2.32	[M + HCOO]^−^	817.49, 655.44, 493.39	Rf3	C_42_H_74_O_15_	818.5028	↑
9	7.05	1007.5419	−1.29	[M + HCOO]^−^	961.54, 799.49, 637.43	Re3	C_48_H_82_O_19_	962.5450	↓
10	7.08	829.4966	1.33	[M + HCOO]^−^	783.49, 637.43, 619.42, 475.38	Isomer of Rg2	C_42_H_72_O_13_	784.4973	↑
11	7.11	683.4384	1.17	[M + HCOO]^−^	637.43, 475.38	F1	C_36_H_62_O_9_	638.4394	↑
12	7.28	1033.5623	3.29	[M + HCOO]^−^	987.56, 945.55, 927.53, 637.43	6′-*O*-acetyl-Re	C_50_H_84_O_19_	988.5607	↓
13	7.34	887.5059	5.52	[M + HCOO]^−^	841.49, 781.48, 437.43	yesanchinoside D	C_44_H_74_O_15_	842.5028	↓
14	8.41	650.3175	3.08	[M + 2HCOO]^2−^	1209.62, 1077.58, 945.54, 783.49, 621.44	Ra1	C_58_H_98_O_26_	1210.6346	↓
15	8.47	683.4381	0.73	[M + HCOO]^−^	637.43, 475.38	Rh1 ^a^	C_36_H_62_O_9_	638.4394	↑
16	8.49	1285.6510	5.91	[M + HCOO]^−^	1107.60, 1077.57, 945.54	R4/Ra3	C_59_H_100_O_27_	1240.6452	↓
17	8.55	1153.6011	0	[M + HCOO]^−^	1107.60, 945.54, 783.49	Rb1 ^a^	C_54_H_92_O_23_	1108.6029	↓
18	8.61	829.4938	−2.05	[M + HCOO]^−^	783.49, 637.43, 475.38	Rg2	C_42_H_72_O_13_	784.4973	↑
19	8.82	1077.5833	−1.67	[M − H]^−^	945.54, 915.53, 783.49	Rc	C_53_H_90_O_22_	1078.5924	↓
20	8.85	650.3211	8.61	[M + 2HCOO]^2−^	1209.63, 1077.58, 945.54, 783.49, 621.44	Ra2	C_58_H_98_O_26_	1210.6346	↓
21	9.13	1077.5958	9.93	[M − H]^−^	945.54, 915.53, 783.49	Rb2	C_53_H_90_O_22_	1078.5923	↓
22	9.3	845.4962	6.86	[M + HCOO]^−^	799.48, 637.43, 475.38	Rg7/Ib/majoroside F2	C_42_H_72_O_14_	800.4922	↑
23	9.48	1195.615	2.76	[M + HCOO]^−^	1149.60, 1107.59, 1189.58, 945.55	quinquenoside-R1	C_56_H_94_O_24_	1150.6135	↓
24	9.78	945.5385	−4.55	[M − H]^−^	783.49, 621.44	Rd	C_48_H_82_O_18_	946.5501	↓
25	9.81	605.2981	−2.64	[M + 2HCOO]^2−^	1119.59, 1059.58	Rs1	C_55_H_92_O_23_	1120.6029	↓
26	9.88	845.4927	2.72	[M + HCOO]^−^	799.48, 637.43, 475.38	Rg7/Ib/majoroside F2	C_42_H_72_O_14_	800.4922	↑
27	10.1	1165.5989	−1.89	[M + HCOO]^−^	1119.60, 1077.59, 783.49	Rs2	C_55_H_92_O_23_	1120.6029	↓
28	10.23	845.4913	1.06	[M + HCOO]^−^	799.45, 637.43, 475.38	Rg7/Ib/majoroside F2	C_42_H_72_O_14_	800.4922	↑
29	10.27	991.5427	−5.65	[M + HCOO]^−^	945.54, 783.48, 621.43, 459.38	gypenoside-XVII /chikusetsusaponin-FK7	C_48_H_82_O_18_	946.5501	↓
30	10.66	827.4786	−1.45	[M + HCOO]^−^	781.47, 619.42	Rg8	C_42_H_70_O_13_	782.4816	↑
31	10.76	829.5001	5.55	[M + HCOO]^−^	783.49, 621.44, 475.38	Isomer of Rg2	C_42_H_72_O_13_	784.4973	↑
32	10.91	961.539	1.25	[M + HCOO]^−^	915.53, 783.49, 621.43, 459.38	vina-R16/gypenoside-IX /noto-Fe/noto-La/quinquenoside L10	C_47_H_80_O_17_	916.5396	↓
33	10.92	827.4819	2.54	[M + HCOO]^−^	781.47, 619.42	Rg9	C_42_H_70_O_13_	782.4816	↑
34	11.12	829.4992	4.46	[M + HCOO]^−^	783.48, 621.44	F2	C_42_H_72_O_13_	784.4973	↑
35	11.37	797.4692	−0.13	[M + HCOO]^−^	751.46, 619.42, 457.37	Isomer of noto-T5	C_41_H_68_O_12_	752.4711	↑
36	11.69	943.5249	−2.44	[M − H]^−^	665.43, 619.42	(L-1/2/5)-(Glc-Rha)-Glc	C_48_H_80_O_18_	944.5345	↑
37	12.03	829.4985	3.62	[M + HCOO]^−^	783.49, 621.44, 459.38	Rg3 ^a^	C_42_H_72_O_13_	784.4973	↑
38	12.19	521.3816	−6.14	[M + HCOO]^−^	-	Protopanaxatriol	C_30_H_52_O_4_	476.3866	↑
39	12.47	631.3824	−4.43	[M − H]^−^	555.37, 455.35	(I-1)-GlurA	C_36_H_56_O_9_	632.3924	↑
40	12.55	871.5073	1.38	[M + HCOO]^−^	825.50, 783.49, 621.44, 459.38	Rs3	C_44_H_74_O_14_	826.5079	↑
41	12.8	809.4991	−8.15	[M − H]^−^	763.46, 601.41	(F-1/2/3)-(Glc-Glc)-acetyl	C_44_H_74_O_13_	810.5129	↑
42	13.1	667.4390	−5.54	[M + HCOO]^−^	621.44, 459.38	Rh2 ^a^	C_36_H_62_O_8_	622.4445	↑
43	14.07	649.4282	−6.01	[M + HCOO]^−^	603.42, 441.37	Rh3	C_36_H_60_O_7_	604.4339	↑
44	14.16	953.4875	−9.23	[M + HCOO]^−^	907.57, 783.49, 621.43, 161.05, 141.09	438-(Glc-Rha)-Glc	C_44_H_76_O_19_	908.4981	↑
45	14.75	505.3886	−2.37	[M + HCOO]^−^	-	Protopanaxadiol	C_30_H_52_O_3_	460.3916	↑

^a^ The identification was confirmed by standard. ↓ The intensity of compound in uncompleted black ginseng is more than in completed black ginseng; ↑ the intensity of compound in uncompleted black ginseng is less than in completed black ginseng.

## Data Availability

The data presented in this study are available upon request from the corresponding author.

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
