# Peer review of "Chemical Differentiation and Quantitative Analysis of Black Ginseng Based on an LC-MS Combined with Multivariate Statistical Analysis Approach"

_molecules, 2023, doi:10.3390/molecules28135251_

Round 1

Reviewer 1 Report

The manuscript overall is fine, but some issues. In section 2.1, there are no sentences and citations describing figs 1 b and c. In section 2.4, lines 183-5, there is a very general conclusion statement about pharmacological effects with no citations. This sentence needs a number of references to support it, such as a series of references of medical studies related to the benefits of the ginsenosides that are increased by processing (Rg3, Rh1, Rh2...). Also there needs another section (2.5) added after the discussion for conclusions which would give specific conclusions with names of compounds to summarize major changes in ginsenosides during processing, which ginsenosides compounds are the best markers and why, how chemical reactions during heating could increase or decrease specific ginsenosides based on the chemical reactions, such as acetylation.

Review the English to not start sentences with "and", or use "discussions", "on the contrary", etc...

Reviewer 2 Report

This research article effectively demonstrates the chemical differentiation and quantitative analysis of black ginseng through the implementation of two types of LC-MS strategies. Although the experimental design is sound and the main text is well-crafted, I have a few minor suggestions regarding the figure format and the experimental details.

1. P.3 Figure 1, why there is a difference in the intensities of Rh1 and Rg3 between Figure A and Figure C, where they are approximately 5E4 vs. 3E5? It would greatly improve the comparison if these three graphs were normalized to the same scale.

2. P.4 line 94, “The precisions for all six ginsenosides…”, it would be helpful to provide a more precise definition of "precision". Precision is a multifaceted concept that encompasses intermediate precision and reproducibility, in addition to the repeatability also mentioned by the authors.

3. P.5 line 134, may I know the reason behind the switch of instruments from QqQ to Q-TOF?

4. P.5 line 137, it would be beneficial to have the three graphs normalized to the same scale and accompanied by an overlaid figure as a supplement to the existing stack figure, as it was mentioned that Figure 2 would be comparing the relative height.

5. P.12 line 242, may I know whether authors explored the inter-day precision? I may have overlooked its description in the main text.
